# Floquet non-Abelian topological insulator and multifold bulk-edge correspondence

Tianyu Li ◉[1] & Haiping Hu ◉[1,2] ✉

Topological phases characterized by non-Abelian charges are beyond the scope of the paradigmatic tenfold way and have gained increasing attention recently. Here we investigate topological insulators with multiple tangled gaps in Floquet settings and identify uncharted Floquet non-Abelian topological insulators without any static or Abelian analog. We demonstrate that the bulk-edge correspondence is multifold and follows the multiplication rule of the quaternion group $Q_8$. The same quaternion charge corresponds to several distinct edge-state configurations that are fully determined by phase-band singularities of the time evolution. In the anomalous non-Abelian phase, edge states appear in all bandgaps despite trivial quaternion charge. Furthermore, we uncover an exotic swap effect—the emergence of interface modes with swapped driving, which is a signature of the non-Abelian dynamics and absent in Floquet Abelian systems. Our work, for the first time, presents Floquet topological insulators characterized by non-Abelian charges and opens up exciting possibilities for exploring the rich and uncharted territory of non-equilibrium topological phases.

The past few decades have witnessed a remarkable surge of research in topological phases of matter[1,2], culminating in the renowned Altland-Zirnbauer tenfold way[3–7]. Based on the underlying symmetries and spatial dimensions, gapped bulk Hamiltonians are characterized by Abelian-type topological invariants ($\mathbb{Z}$ or $\mathbb{Z}_2$) with their own manifestation of protected boundary states. Very recently, the notion of band topology has been extended to tangled multi-gap scenarios[8–15]. For instance, in the presence of space-time inversion (PT) symmetry, one-dimensional (1D) insulators involving multiple bandgaps may carry non-Abelian quaternion charges[8] and host richer topological phases as experimentally observed in transmission line networks[16,17]. Yet in its infancy, the tangled multi-gap topology plays a vital role in describing, e.g., the disclination defects of nematic liquids[18–22], the admissible nodal lines[23–27], and the reciprocal braiding of Dirac/Weyl/exceptional points[11,28–31].

Floquet engineering provides a powerful knob in manipulating band structures[32–41], offering unprecedented control over the topological properties of materials and the exploration of non-equilibrium phenomena. With a time-periodic Hamiltonian $H(t) = H(t + T)$ ($T$ is the driving period), the stroboscopic dynamics is dictated by an effective Floquet Hamiltonian. Notably, Floquet systems exhibit intriguing topological features with no static analog arising from the replicas of quasienergy bands, such as the emergence of anomalous chiral edge modes[42–45] despite the triviality of all bulk bands. Incorporating the multi-gap scenario, this paper aims to address three fundamental questions regarding Floquet multi-gap topology. (i) Does a Floquet topological insulating phase characterized by non-Abelian charge exist, and if so, how can it be implemented through periodic driving? (ii) What novel bulk-edge correspondence does such a non-Abelian phase possess, and how can it be described? (iii) Are there any unique topological or dynamical phenomena associated with this phase?

Here we answer these questions affirmatively. Firstly, we propose the realization of the simplest Floquet non-Abelian topological insulator (FNATI) in a 1D three-band system with PT symmetry. Secondly, the FNATI is characterized by a quaternion charge, which, on its own, cannot predict the existence or the number of edge states. Moreover, each quaternion charge corresponds to multiple edge-state configurations, demonstrating a multifold bulk-edge correspondence that

[1]Beijing National Laboratory for Condensed Matter Physics, Institute of Physics, Chinese Academy of Sciences, 100190 Beijing, China. [2]School of Physical Sciences, University of Chinese Academy of Sciences, 100049 Beijing, China. ✉e-mail: hhu@iphy.ac.cn

obeys the multiplication rule of the quaternion group. The full topology or edge-state configuration is completely captured by the phase-band singularities of the time-evolution operator over one driving period. Intriguingly, we identify an anomalous FNATI hosting edge modes inside all bandgaps with a trivial bulk quaternion charge. Thirdly, we reveal the emergence of interface modes with swapped driving sequences as a genuine non-Abelian effect. It indicates the non-commutative nature of the FNATI. This is in sharp contrast to Floquet Abelian topological insulators, where such interface modes are absent due to the same spectral structures regardless of the choice of time frame. We emphasize that the intriguing properties of FNATI stem from the presence of multiple tangled bandgaps. Our findings expand the scope of Floquet topological insulators into the non-Abelian regime and open up new avenues for investigating the vast and unexplored territory of non-equilibrium topological phases.

## Results

### Multi-gap topology and driving protocol

Let us recap the static three-band topological insulator characterized by the quaternion charge $Q_8$[8]. In the presence of PT symmetry, the Hamiltonian becomes real-valued in momentum space $H(k) = H^*(k)$ when expressed on a suitable basis. Consequently, the eigenstates represent three real vectors that are orthonormal to each other and span a coordinate frame, as sketched in Fig. 1a. When simultaneously considering both bandgaps, the configuration space of the Hamiltonian is $M_3 = \frac{O(3)}{O(1)^3}$, where $O(N)$ denotes the orthogonal group of $N$ dimension. The $O(1) = \mathbb{Z}_2$ factor comes from the gauge freedom ($\pm 1$) for each real eigenstate. With the variation of momentum $k$ from $-\pi$ to $\pi$, the eigenstate frame rotates on the unit sphere. The mapping from the 1D Brillouin zone $k \in [-\pi, \pi] = S^1$ to $M_3$ is governed by the fundamental group of $M_3$, which describes the frame rotation. This fundamental group is given by the quaternion group $\pi_1(M_3) = Q_8$[8]. As a non-Abelian group, $Q_8$ has eight elements and five conjugacy classes $\{1, \pm i, \pm j, \pm k, -1\}$ with multiplication rule $i^2 = j^2 = k^2 = ijk = -1$, $ij = -ji$, $ik = -ki$, $jk = -kj$. The quaternion charge captures the multi-gap band topology and governs the number of edge states in both bandgaps[16]. The trivial phase with charge $q = 1$ has no edge states under open boundaries.

We consider a 1D lattice with three sites, denoted as $A, B, C$ per unit cell. Our Floquet driving is based on two ingredients, namely $H_1$ and $H_2$ as depicted in Fig. 1b. $H_1$ ($H_2$) contains only intracell (intercell) couplings and respects PT symmetry,

$$\begin{cases} H_1 = \sum_{n=1}^{L} \sum_{X,Y} s_{XY} c_{X,n}^\dagger c_{Y,n} + h.c., \\ H_2 = \sum_{n=1}^{L-1} \sum_{X,Y} v_{XY} c_{X,n}^\dagger c_{Y,n+1} + h.c.. \end{cases} \quad (1)$$

Here $c_{X,n}^\dagger$ and $c_{X,n}$ represent the creation and annihilation operators at site $X$ ($X = A, B, C$) of the $n$-th unit cell, respectively. The lattice length is $L$. The coupling parameters $s_{XY}$ and $v_{XY}$ used in this paper are listed in the Methods. Without loss of generality, the driving period is set as $T = 1$. We adopt a symmetric driving protocol: $H(t) = H_1$ for $t \in mT + [0, T/4] \cup [3T/4, T]$ and $H(t) = H_2$ for $t \in mT + [T/4, 3T/4]$ ($m \in \mathbb{Z}$). The dynamics of the system is governed by the time evolution operator $U(t) = \mathcal{T} e^{-i \int_0^t H(\tau) d\tau}$, where $\mathcal{T}$ means time ordering. The stroboscopic evolution of the system is described by the Floquet operator,

$$U(T) = e^{-iH_1 T/4} e^{-iH_2 T/2} e^{-iH_1 T/4}. \quad (2)$$

The effective Floquet Hamiltonian $H_F$ is defined through $U(T) = e^{-iH_F T}$. Due to the symmetric driving, the Floquet Hamiltonian respects PT symmetry, i.e., $H_F(k) = H_F^*(k)$. After the diagonalization: $H_F |u_n\rangle = \epsilon_n |u_n\rangle$ ($n$ is the band index), we obtain the quasienergy $\epsilon_n$ and eigenstates $|u_n\rangle$. The quasienergies are well-defined modulo $2\pi/T$, and

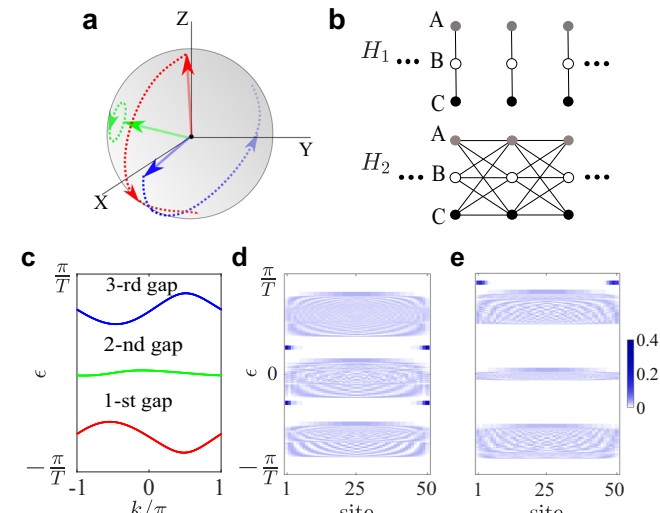

**Fig. 1 | Multi-gap topology in Floquet systems. a** Sketch of the frame rotation of eigenstates for topological insulator characterized by quaternion charge. The loci of eigenstate on the unit sphere with varying momentum $k$ from $-\pi$ to $\pi$ are marked in red/green/blue for the first/second/third band, respectively. **b** Building blocks for our Floquet driving. $H_1$ ($H_2$) contains only intracell (intercell) coupling terms. **c** Labeling of the quasienergy bandgaps. The third gap straddles the FBZ edge at $\pm \frac{\pi}{T}$. **d, e** Quasienergy spectra and spatial distributions (represented by color shades) of their eigenstates for quaternion charge $q = j$ with open boundaries. In **d**, the edge states emerge at both the first and second gaps. In **e** the edge states emerge at the third gap. The lattice length is $L = 50$. The parameters are listed in "Methods" section.

form the quasienergy bands. In the following, we set $\epsilon_n$ to be in the first Brillouin zone (FBZ) of quasienergy $\epsilon_n \in (-\pi/T, \pi/T)$.

Unlike the static three-band model with two bandgaps, the Floquet system exhibits an additional bandgap that spans across the FBZ boundary at $\pm \pi/T$ as shown in Fig. 1c, which arises from the replica of Floquet spectra. As a consequence, the multi-gap topology is greatly enriched in Floquet settings. The closing and reopening of this bandgap lead to the emergence of extra edge modes under open boundary conditions. For ease of reference, we denote the band from bottom to top as the first, second, and third bands, respectively, while bearing in mind their replicated nature. The gap between them is then labeled as the first, second, and third bandgap. In Fig. 1d, e, we present the quasienergy spectra and their spatial profiles (of their associated eigenstates) with open boundaries. In both cases, the bulk Floquet Hamiltonian carries a quaternion charge of $q = j$. In Fig. 1d, we observe one edge mode (for each end of the lattice) at the first and second bandgaps, respectively, which is similar to the static case[16]. While in Fig. 1e, there is only a single edge mode within the third bandgap. Similar scenarios apply to other cases as well. For instance, for charge $i$, edge states can exist in both the first and third bandgaps or solely in the second gap. Likewise, for charge $k$, the edge state may exist in both the second and third bandgaps or only in the first gap.

### Anomalous FNATI

Owning to the additional bandgap at the FBZ edge, the periodically driven system can exhibit an intriguing phase with protected edge modes even when the bulk quaternion invariant is trivial. This is in stark contrast with the static systems[16], wherein the charge $q = 1$ system does not possess any nontrivial edge states. Note that a similar anomaly also happens on 2D driven lattice with protected chiral edge states despite the triviality of bulk Chern bands[42]. We thus dub this phase anomalous FNATI. In Fig. 2a, we plot the quasienergies and the spatial profiles of their associated eigenstates for this anomalous

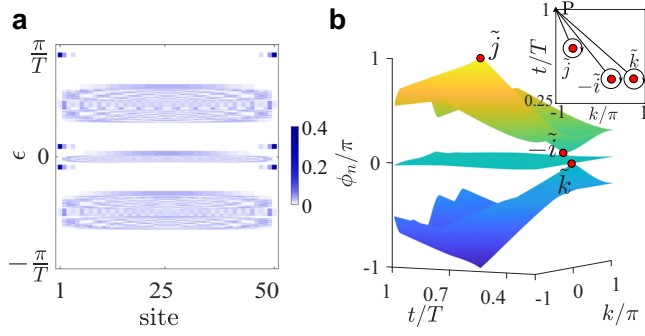

**Fig. 2 | FNATI with anomalous edge states. a** Quasienergy spectra and the spatial distributions of their eigenstates with open boundaries. The bulk quaternion charge is $q = 1$. The lattice length is $L = 50$. **b** Phase bands of $\tilde{U}(k, t)$ in the 2D momentum-time space. The Dirac-point singularities are marked with red dots. The inset: locations of the Dirac points with quaternion charges $\tilde{j}$, $-\tilde{i}$, and $\tilde{k}$ in the $(k, t)$-space. $P$ is the base point for the encircling paths. The quaternion charge of the bulk Floquet Hamiltonian and Dirac points are related via $1 = \tilde{k} \cdot (-\tilde{i}) \cdot \tilde{j}$. The parameters are listed in "Methods" section.

phase. We can clearly observe the existence of edge states (one for each end of the lattice) in all three bandgaps.

In the anomalous phase, the bulk invariant of the Floquet Hamiltonian, regardless of whether it is the quaternion charge or Berry phase of Abelian type introduced for static systems[16], is unable to view the edge-state landscape. In fact, the emergence of edge state has dynamical origin and the complete information of the dynamical topology is encoded in the full-time evolution operator. A unified analysis of the bulk-edge correspondence can be achieved through the introduction of phase band and its associated momentum-time singularities[41,46-48]. It should be noted that the time-evolution operator $U(t)$ is not always PT symmetric. To this end, we adopt a PT symmetric operator $\tilde{U}(k, t)$ via a smooth deformation of $U(t)$ (see "Methods" section). $\tilde{U}(k, t)$ preserves the phase-band structure (including its singularities) and satisfies $\tilde{U}(k, t) = U(k, t)$ at $t = 0, T$. Formally, the operator $\tilde{U}(k, t)$ can be expressed in terms of the spectral decomposition,

$$\tilde{U}(k, t) = \sum_{n=1}^{3} e^{-i\phi_n(k,t)} |\psi_n(k, t)\rangle \langle \psi_n(k, t)|, \qquad (3)$$

where $e^{-i\phi_n(k,t)}$ denotes the eigenvalues of $\tilde{U}(k, t)$ and $\phi_n(k, t) \in (-\pi/T, \pi/T]$ forms the phase bands in the 2D momentum-time space.

The phase band may touch and reopen with the variation of time, leaving degenerate Dirac points in the bandgaps in the 2D momentum-time space. At $t = T$, the phase bands become the quasienergy bands. As shown in Fig. 2b, the phase bands touch once in each gap for the anomalous phase. The presence of Dirac singularities in the phase band leads to the emergence of edge states in the corresponding gap. As the three gaps are tangled, each Dirac point can be assigned a quaternion charge through the Wilson loop along an enclosing path nearby (See Supplementary Material Appendix A). The quaternion charge $q$ of the Floquet Hamiltonian $H_F$ is related to the quaternion charge of the phase-band singularities via

$$q = \prod_m \tilde{q}_m. \qquad (4)$$

Here $m$ labels the Dirac point, and the multiplication is ordered such that the concatenation of the small enclosing paths coincide with the orientation of momentum integral $-\pi \to \pi$. As a distinction, the charge of the Dirac singularity is marked with a tilde. In the anomalous FNATI depicted in Fig. 2b, the three singularities from right to left possess

quaternion charge $\tilde{q}_1 = \tilde{k}, \tilde{q}_2 = -\tilde{i}$, and $\tilde{q}_3 = \tilde{j}$, respectively [See the inset]. It is evident that the relation $1 = \tilde{k} \cdot (-\tilde{i}) \cdot \tilde{j}$ holds.

## Multifold bulk-edge correspondence

The phase-band singularities and their non-Abelian nature offer a straightforward interpretation of the bulk-edge correspondence in FNATI. As indicated by Eq. (4), various patterns of edge states may correspond to the same quaternion charge of the Floquet Hamiltonian. Such multifold bulk-edge correspondence follows the multiplication rule of the quaternion group $Q_8$. For example, in Fig. 1d,e, the two phases with the same quaternion charge $q = j$ have distinct edge-state patterns. In their phase bands, there are two Dirac singularities with charge $\tilde{k}$ and $\tilde{i}$ for the former, while there is a single singularity with charge $\tilde{j}$ for the latter. They satisfy the multiplication rule $\tilde{k} \cdot \tilde{i} = \tilde{j}$. Similar discussions can be applied to other cases. The multifold bulk-edge correspondence can be deduced from two key observations. Firstly, the edge state within each bandgap is determined by the phase-band closings that occur within that bandgap during time evolution. Each gap closing results in a change of the mass term and the emergence of an in-gap mode by Jackiw-Rebbi's argument[49]. With multiple bandgaps present, the edge-state patterns are determined by considering the phase-band singularities within each bandgap. Secondly, the phase-band singularities relate to the bulk quaternion charge via Eq. (4). It then becomes evident that the edge-state pattern is linked to the quaternion charge, and the bulk-edge correspondence showcases a multifold nature that adheres to the multiplication rule.

In Fig. 3, we present a comprehensive list of bandgap closings and their corresponding edge-state patterns (per edge) for all quaternion charges. Notably, we observe similar bandgap closings and edge-state configurations for quaternion charges that belong to the same conjugacy class, such as $\pm i$. This is due to the anti-commutating relations of quaternion charge. For example, the existence of a type $\tilde{j}$ and type $\tilde{k}$ singularity in the phase band (regardless of their ordering) results in two edge states (located at the first and third bandgap). Switching them yields conjugate quaternion charges: $\tilde{j} \cdot \tilde{k} = -\tilde{k} \cdot \tilde{j}$. In addition to the patterns that appeared in the static case (the fourth column)[16], Floquet systems can exhibit unique edge-state patterns (the fifth column) because of the additional phase-band touchings at the FBZ edge. It should be noted that for a particular gap, multiple touchings can exist (See Supplementary Material Appendix B), such as $i = \tilde{k} \cdot (-\tilde{k}) \cdot \tilde{i}$, which has two Dirac points in the first bandgap. However, such cases can be reduced to the simpler cases listed in Fig. 3 by eliminating two Dirac points of the same gap pairwise, resulting in a factor $\pm 1$. Furthermore, in the case of charge $q = -1$, two fickle edge states[16] (with an unspecified bandgap) may appear according to $-1 = \tilde{i}^2 = \tilde{j}^2 = \tilde{k}^2$, or three separate edge states may appear according to $-1 = \tilde{i} \cdot \tilde{j} \cdot \tilde{k}$. We note that the multifold bulk-edge correspondence can be extended directly to the domain-wall problem. For two Floquet systems with charge $q_L$ and $q_R$, the patterns of topological domain-wall states between them are dictated by the quotient $q_L / q_R$. Different from the static case, the bandgap closings across the domain wall are fully captured by the phase-band singularities, yielding a multifold bulk-domain-wall correspondence (See Supplementary Material Appendix D, E). Last, our numerical analysis indicates that the edge/domain-wall states are robust against small disorders. In the presence of domain-wall decorations, it is possible for additional trivial bound states to emerge. However, the non-trivial domain states, governed by the non-Abelian topology, remain unaffected (See Supplementary Material Appendix F).

## Interface modes induced by swapped driving

Besides the multifold bulk-edge correspondence, here we uncover a counter-intuitive phenomenon of FNATI as a manifestation of the

| Quaternion charge | Phase-band crossings | | Edge states |
|---|---|---|---|
| *1* | | | |
| $\{\pm i\}$ | | | |
| $\{\pm j\}$ | | | |
| $\{\pm k\}$ | | | |
| *-1* | | ... | ... |

**Fig. 3 | Multifold bulk-edge correspondence for FNATI.** The first column lists the quaternion charge for the Floquet Hamiltonian, and the five conjugacy classes of $Q_8$ correspond to different rows. The second and third columns sketch the phase band structure with singularities depicted by red dots. The cases with band crossing at the FBZ edge are listed in the third column. The fourth/fifth columns sketches the edge-state patterns corresponding to the phase bands in the second/third column. The black dots represent edge states. Empty circles mark the fickle edge states. The configurations unique to Floquet systems are listed in fifth column. The list applies equally to the domain-wall problem.

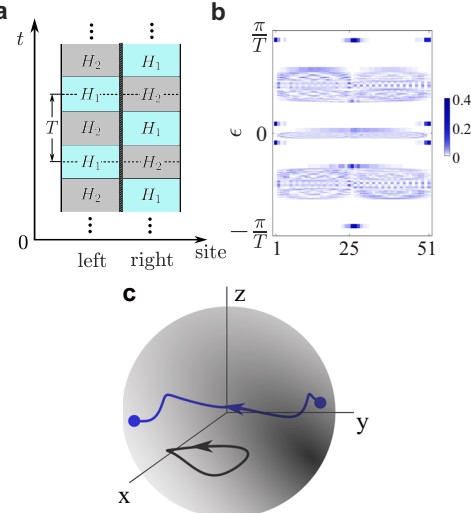

**Fig. 4 | Interface modes with swapped driving as a manifestation of the non-Abelian topology. a** The Floquet setting: two subsystems connected by an interface have swapped driving sequences. **b** Quasienergy spectra and spatial profiles of their eigenstates with open boundaries at the two ends. A pair of domain-wall states emerge at the third bandgap. The lattice length is $L = 51$, and the interface is located at $n = 26$. Other parameters are the same as Fig. 2. **c** Trajectory of the transformation $O$ (blue curve) in the parameter space of $SO(3)$ (represented by a solid ball). As $k$ varies from $-\pi$ to $\pi$, $O$ follows a closed nontrivial loop that connects a pair of antipodal points in $SO(3)$. For comparison, the black curve represents a trivial loop in $SO(3)$.

**Floquet non-Abelian topology.** As illustrated in Fig. 4a, we consider a system with two sides featuring swapped (mismatched) driving sequences. Within one full period, the Hamiltonians on the left and right sides are given by $H_1 \to H_2 \to H_1$ and $H_2 \to H_1 \to H_2$, respectively. The bulk Floquet operators for these two sides, denoted as $U_L$ and $U_R$, are related via a similarity transformation $U_L = V^{-1} U_R V$ where $V = e^{-iH_2 T/4} e^{-iH_1 T/4}$ accounts for the time shift. This shift does not alter the quasienergy spectra irrespective of the boundary conditions. In Floquet Abelian topological phases[39,40], the bulk topological invariant is defined for each individual quasienergy bandgap (or branch-cut). The two subsystems should have the same topological invariant for each bandgap to match with the edge states, and no stable interface modes are expected to exist. However, in our non-Abelian system with multiple intertwined gaps, this is not always true. In Fig. 4b, we plot the quasienergy spectra in the presence of an interface of swapped driving using the same parameters as Fig. 2. We can clearly observe a pair of domain-wall states (located near $\pm \frac{\pi}{T}$) in the third gap. The emergence of the interface modes signifies the non-commutativity of the two driving sequences and it is a genuine non-Abelian effect of the Floquet dynamics. We have also verified the appearance of domain-wall states for other cases with different quaternion charges.

To understand this fascinating effect, we define the bulk eigenstates of $U_L$ and $U_R$ on two sides as $S_L = (|u_{1L}\rangle, |u_{2L}\rangle, |u_{3L}\rangle)$ and $S_R = (|u_{1R}\rangle, |u_{2R}\rangle, |u_{3R}\rangle)$. With PT symmetry, they are related by an $SO(3)$ transformation $S_L = OS_R$, with $O = S_L S_R^{\mathbb{T}}$ ($\mathbb{T}$ denotes transposition). It is important to note that $O$ may alter the quaternion charge. In particular, when $O$ corresponds to the non-identity element in the fundamental group $\pi_1(SO(3)) = \mathbb{Z}_2$, the bulk quaternion charge for the two sides satisfy $q_L = -q_R$. In fact, according to the exact sequence of homotopy groups[50]:

$$\cdots \to \pi_1(SO(3)) \xrightarrow{j_1} \pi_1\left(\frac{SO(3)}{D_2}\right) \xrightarrow{\partial_1} \pi_0(D_2) i_0 0, \quad (5)$$

the kernel of $\partial_1$-mapping is $\{1, -1\}$ in $\pi_1(\frac{SO(3)}{D_2}) = Q_8$. Thus $j_1$ maps the non-identity element of $\mathbb{Z}_2$ to $-1$ in $Q_8$. For our case, $O$ traces a nontrivial path in $SO(3)$ as the momentum $k$ varies from $-\pi$ to $\pi$ as depicted in Fig. 4c. $U_L$ has a quaternion charge of $q_L = 1$, while $U_R$ has $q_R = -1$, indicating the emergence of domain-wall states. Alternatively, one can scrutinize this effect through the phase-band singularities (See Supplementary Material Appendix C). With time variation, the phase band undergoes crossings with charge $\tilde{k}, -\tilde{i}, \tilde{j}$ and $-\tilde{k}, \tilde{i}, -\tilde{j}$ for $U_L$ and $U_R$, respectively. They satisfy Eq. (4), $q_L = 1 = \tilde{k} \cdot (-\tilde{i}) \cdot \tilde{j}$ and $q_R = -1 = (-\tilde{k}) \cdot \tilde{i} \cdot (-\tilde{j})$.

## Discussion

In summary, our exploration of the FNATI represents a significant advancement in the field of out-of-equilibrium topological phases. We have demonstrated the implementation of FNATI through step-like driving and fully characterized its topological properties via phase-band singularities. Our analysis revealed a multifold bulk-edge correspondence, governed by the multiplication rule of the quaternion group $Q_8$. We have identified an anomalous phase that possesses edge modes within all bandgaps despite a trivial bulk, which lacks a static analog. Furthermore, we have uncovered a novel interface effect resulting from swapped driving that serves as a key signature of the non-Abelian topology.

Our findings offer novel insights into Floquet topological phases with multiple intertwined bandgaps. Our results can be extended to higher-band cases, such as a four-band topological insulator characterized by the non-Abelian group $Q_{16}$[17]. In higher dimensions, the phase-band singularities may extend to nodal lines, and it would be intriguing to explore their interwinding in momentum-time space and the associated non-Abelian effects. From a wider perspective, the multi-gap topology belongs to the fragile topology and relies on the partition of energy bands[10]. A different partition $(2 + 1)$ yields a different flag manifold $RP^2$ and is relevant for the Floquet Euler phase[14]. Other tantalizing extensions include the 3D nodal-line metals[8] characterized by non-Abelian

frame charges and non-Hermitian phases with non-Abelian band braidings[51–53] by relaxing the PT symmetry to allow for a complex flag manifold. A natural issue is how these non-Abelian features interplay with Floquet driving. Besides the step-like driving utilized for illustration purposes, smooth Floquet driving should also be suitable for implementing FNATI. With the high feasibility of Floquet engineering[32–41] and the ability to manipulate tight-binding models in various platforms, such as ultracold atoms[45,54–57], photonic or acoustic materials[43,44,58,59], we anticipate that the uncharted FNATI and non-Abelian effect will be observed in near-future experiments.

## Methods

### Calculation of quaternion charge

The quaternion charge $q \in Q_8$ characterizes the rotations of the eigenstate frame as the momentum $k$ varies from $-\pi$ to $\pi$. In the case of the Dirac singularity in the phase band, the quaternion charge describes the frame rotation along the enclosing path. According to ref. [8], the generalized Wilson operator can be used to obtain the quaternion charge. Formally, the Wilson loop along a closed path $\Gamma$ is defined as follows:

$$W_\Gamma = \mathcal{P}e^{\oint_\Gamma A_{\mathrm{all}}(k)\cdot dk}. \tag{6}$$

Here, $[A_{\mathrm{all}}(k)]_{mn} = \langle u_m(k)|\partial_k|u_n(k)\rangle$ represents the Berry-Wilczek-Zee connection. The band indices $m$ and $n$ take values from 1 to 3, and $|u_n(k)\rangle$ is the eigenstate of the Floquet Hamiltonian $H_F$. For the Dirac-point case, we use the phase band instead. $A_{\mathrm{all}}(k)$ is anti-symmetric and can be decomposed into the $\mathfrak{so}(3)$ Lie-algebra basis:

$$A_{\mathrm{all}}(k) = \sum_{i=1,2,3} \beta_i L_i, \tag{7}$$

where $(L_i)_{jk} = -\epsilon_{ijk}$ and $\epsilon_{ijk}$ is the anti-symmetric tensor. We then lift $A_{\mathrm{all}}$ to the $\mathfrak{spin}(3)$-valued 1-form[8] by replacing $L_i$ with $t_i$, where $t_i = -\frac{i}{2}\sigma_i$ and $\sigma_i$ represents the Pauli matrix. This gives us:

$$\overline{A}_{\mathrm{all}}(k) = \sum_{i=1,2,3} \beta_i t_i. \tag{8}$$

Finally, the non-Abelian charge is defined by:

$$q = \mathcal{P}e^{\oint_\Gamma \overline{A}_{\mathrm{all}}(k)\cdot dk}. \tag{9}$$

The elements of the quaternion group are represented by $1 \to \sigma_0$, $i \to -i\sigma_x$, $j \to -i\sigma_y$, and $k \to -i\sigma_z$.

### Smooth deformation of time evolution

To define the quaternion charge of the Dirac singularity, real wave functions of the phase bands are required. To this end, we smoothly deform $U(k,t)$ into $\tilde{U}(k,t)$ such that $\tilde{U}(k,t)U(k,t) = 1$ for all $t \in [0,1]$ (we set $T = 1$ for convenience). The time evolution operator for our driving protocol is

$$U = \begin{cases} e^{-iH_1 t}, & t \in [0,1/4], \\ e^{-iH_2(t-1/4)}e^{-iH_1/4}, & t \in [1/4,3/4], \\ e^{-iH_1(t-3/4)}e^{-iH_2/2}e^{-iH_1/4}, & t \in [3/4,1]. \end{cases} \tag{10}$$

We can define the PT symmetric operator $\tilde{U}(k,t)$ as

$$\tilde{U} = \begin{cases} e^{-iH_1 t}, & t \in [0,1/4], \\ e^{-iH_1/8}e^{-iH_2(t-1/4)}e^{-iH_1/8}, & t \in [1/4,3/4], \\ e^{-\frac{i}{2}(t-\frac{1}{2})H_1}e^{-iH_2/2}e^{-\frac{i}{2}(t-\frac{1}{2})H_1}, & t \in [3/4,1]. \end{cases} \tag{11}$$

## Table 1 | Coefficients used in the paper

| $q$ | $r$ | $s$ | $t$ | $u$ | $v$ | $w$ | $v_{AA}$ | $v_{BB}$ | $v_{CC}$ |
|---|---|---|---|---|---|---|---|---|---|
| $j$ [Fig. 1d] | 1 | 1 | 2 | 2 | 0 | 0 | 0 | –2 | 2 |
| $j$ [Fig. 1e] | 1 | 0 | 4 | 3 | 0 | 0 | –1 | 0 | 1 |
| 1 [Fig. 2a] | 1 | 0 | 3 | 0 | 0 | –3 | –2 | 0 | 2 |
| $i$ [Fig. S2a] | 1 | 0 | 1 | 0 | 1 | 1 | –1 | 0 | 1 |
| –1 [Fig. S2b] | 1 | 0 | 1 | –1 | 0 | 1 | 0 | –2 | 2 |
| $j$ [Fig. S4a,b(left)] | 1 | 0 | 4 | 3 | 0 | 0 | –1 | 0 | 1 |
| $j$ [Fig. S4a(right)] | 1 | 1 | 2 | 2 | 0 | 0 | 0 | -2 | 2 |
| $j$ [Fig. S4b(right)] | 1 | 0 | 4 | 1.4 | 0 | 1.4 | –2 | 0 | 2 |
| $i$ [Fig. S5–S7] | 1 | 0 | 2 | 0 | 1 | 1 | –2 | 0 | 2 |
| $k$ [Fig. S5–S7] | 1 | 1 | 1 | –1 | 1 | 0 | 0 | –1 | 1 |

To visualize the smoothness of the deformation, let us consider the continuous interpolation between them:

$$\mathcal{U}(s,k,t) = \begin{cases} e^{-iH_1 t}, & t \in [0,1/4], \\ e^{-iH_1\frac{s}{8}}e^{-iH_2(t-1/4)}e^{-iH_1(1-\frac{s}{2})\frac{1}{4}}, & t \in [1/4,3/4], \\ e^{-i[(1-\frac{s}{2})t+\frac{2s-3}{4}]H_1}e^{-iH_2/2}e^{-i(\frac{s}{2}t+\frac{1-2s}{4})H_1}, & t \in [3/4,1], \end{cases} \tag{12}$$

such that $\mathcal{U}(s=0,k,t) = U(k,t)$ and $\mathcal{U}(s=1,k,t) = \tilde{U}(k,t)$. During the deformation from $s = 0$ to $s = 1$, the phase bands and the Dirac singularities are unchanged.

### Coefficients of the tight binding model

The coefficients used in the paper are listed in Table 1. In all the cases, we take $s_{AA} = s_{BB} = s_{CC} = 0$, $s_{AB} = s_{BA} = r$, $s_{BC} = s_{CB} = s$, $s_{CA} = s_{AC} = t$ and $v_{AB} = v_{BA} = iu$, $v_{BC} = v_{CB} = iv$, $v_{CA} = v_{AC} = iw$, where $r$, $s$, $t$, $u$, $v$ and $w$ are all real numbers.

## Data availability

All data is available upon reasonable request.

## Code availability

The code that supports the findings of this study is available at https://doi.org/10.5281/zenodo.8294074.

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

## Acknowledgements

This work is supported by the National Key Research and Development Program of China (Grant No. 2022YFA1405800) and the start-up grant of IOP-CAS. T.L. also acknowledges the support from the Project Funded by China Postdoctoral Science Foundation (Grant No. 2023M733719).

## Author contributions

H.H. conceived the main idea, and performed the theoretical analysis with T.L.; T.L. did the numerical calculations. Both authors contribute to the writing of the paper.

## Competing interests

The authors declare no competing interests.
