## [Peer Review File · Nature Communications]

Floquet Non-Abelian Topological Insulator and Multifold Bulk-Edge CorrespondenceREVIEWER COMMENTS

Reviewer #1 (Remarks to the Author):

In this work, the authors investigate the periodically driven version of the recently discovered multi-gap topological band insulators characterized by a non-Abelian quaternion charge. By designing a driving protocol that respects the necessary PT symmetry, the authors identify an anomalous Floquet non-Abelian topological phase without a static counterpart, due to the periodic nature of the quasienergy spectrum and the additional gap that arises at quasienergy π . The authors further characterize such anomalous phases using phase band singularities.

I find that the paper is overall very clearly written, and accessible to a general audience. However, I am not convinced that the results reported in this work represent a significant advancement in the field of out-of-equilibrium topological phases. The two key ingredients in the phenomena studied in this work, namely, the non-Abelian topological invariant and the phase band singularities, have already been explored. In particular, the notion of anomalous Floquet topological phases and their characterizations in terms of phase band singularities are barely new at this point. In fact, I find that it has already become a paradigm that people can readily apply to whatever new topological band insulators in static systems, and construct its driven version where an "anomalous" phase shows up. Therefore, I do not see a significant conceptual or technical progress in this work to warrant publication in *Nature Communications*.

Reviewer #2 (Remarks to the Author):

The author studied the PT -symmetric periodically driven 3-band model with non-Abelian topological charge, proposed the realization of anomalous Floquet non-Abelian topological insulator phase, and signatures of interface modes induced by swapped driving as a probe of non-Abelian topology.

Overall, the manuscript studied a very interesting problem. However, in the current manuscript the discussion of bulk-boundary correspondence is more like enumerating things without revealing much of deeper understanding. One should be able to provide some argument about why the boundary states should look at the list in Fig. 3, and why the domain wall problem corresponds to the quotient of left side and right side. These are claims in the current version that need some more explanation. I guess some type of Zak phase calculation would apply similar to Ref. 16. But still a good discussion in the Floquet case is needed for better understanding of the manuscript. For this domain wall problem, I have a minor technical question: maybe I have some misunderstanding but why the quotient is -1 for the swapped driving case but the interface states look like j in Fig. 3?

Furthermore, usually for the topology of a fixed number of bands such as the multi-gap topology here, the bulk-boundary correspondence is sensitive to boundary conditions since on boundaries/domain walls there can be more degrees of freedom. This is what happened in e.g., delicate topology. I am wondering how general the bulk boundary correspondence is upon changing boundary condition or decorating boundaries/domain walls and how stable it is about the appearance of domain wall states. I believe here some precise statements need to be made to clarify the stability of the bulk-boundary correspondence especially for the intrinsically non-Abelian part.

In summary, I believe the current manuscript can be improved to meet the standard of Nature Communication and a more transparent and thorough understanding of the phenomena discussed in the current manuscript is needed.

Reviewer #3 (Remarks to the Author):<

This work proposes a Floquet topological insulator model characterized by non-abelian charge. They show the bulk-edge correspondence can be derived from the phase-band singularities, and there is an "anomalous" phase in which the bulk topological invariant is trivial but the edge mode is protected. They also found a novel interface effect as a result of the non-Abelian topology.

The dynamics of non-abelian topological systems is a developing field with many open questions to address. This work provides the first Floquet model with non-Abelian topology and thoroughly studies the bulk-edge correspondence to the best of my knowledge. Their findings of the "anomalous" edge state and the interface modes are also remarkable. These predictions might be tested in future experiments. So far as I see, they provide enough analytical and numerical calculations to justify their statements. Therefore, I would like to recommend its publication on Nat Com.

There are two questions that the authors might want to address in a modified manuscript. 1) Can the authors comment on the possibilities of generalizing their work to higher spatial dimensions, like 2D? 2) Can the authors comment on the possibilities of generalizing their work to Floquet Chern insulators?

Response to the review reports of manuscript NCOMMS-23-16219 “Floquet Non-Abelian Topological Insulator and Multifold Bulk-Edge Correspondence”

We appreciate the valuable comments and suggestions provided by all three reviewers (copied in blue font). We have carefully considered their feedback and incorporated the necessary revisions into the manuscript. Our point-by-point responses are presented in black font. A comprehensive list of the changes made can be found at the end of this document.

Reply to Reviewer #1

[**Comment**] In this work, the authors investigate the periodically driven version of the recently discovered multi-gap topological band insulators characterized by a non-Abelian quaternion charge. By designing a driving protocol that respects the necessary PT symmetry, the authors identify an anomalous Floquet non-Abelian topological phase without a static counterpart, due to the periodic nature of the quasienergy spectrum and the additional gap that arises at quasienergy π . The authors further characterize such anomalous phases using phase band singularities.

[**Reply**] We thank Reviewer #1 for the careful review and accurate description of our results.

[**Comment**] I find that the paper is overall very clearly written, and accessible to a general audience. However, I am not convinced that the results reported in this work represent a significant advancement in the field of out-of-equilibrium topological phases. The two key ingredients in the phenomena studied in this work, namely, the non-Abelian topological invariant and the phase band singularities, have already been explored. In particular, the notion of anomalous Floquet topological phases and their characterizations in terms of phase band singularities are barely new at this point. In fact, I find that it has already become a paradigm that people can readily apply to whatever new topological band insulators in static systems, and construct its driven version where an “anomalous” phase shows up. Therefore, I do not see a significant conceptual or technical progress in this work to warrant publication in Nature Communications.

[**Reply**] We appreciate Review #1’s comment that “*the paper is overall very clearly written, and accessible to a general audience.*” We also agree with the reviewer that the phase-band singularity, which is important in the characterization of Floquet Abelian topological phases,

has been explored in some other contexts, e.g., Floquet Chern insulator. However, we would like to clarify the novelty and impact of our work from the following three aspects.

(1) Firstly, we stress that the phase-band singularities and the non-Abelian topological invariants, are **not the main result of our paper**. They **merely serve as theoretical tools** for our analysis and provide a prelude to our conclusions. Instead, through the analysis of the phase-band singularity and non-Abelian topological invariants, we have obtained a series of new results presented in three sections of our manuscript [*Anomalous FNATI; Multifold bulk-edge correspondence; Interface modes induced by swapped driving*], which are the main part of our paper. They include: (i) We have identified a series of Floquet non-Abelian topological insulators (FNATIs), including the anomalous phase, together with a comprehensive classification and concrete model proposals. (ii) We have shown that the edge/domain-wall states in FNATIs exhibit a novel multifold bulk-edge correspondence that follows the multiplication rules of the quaternion group, which is entirely different from the Abelian case in the literature. (iii) We have uncovered a unique interface effect specific to FNATIs, without any static or Abelian counterpart.

(2) Floquet topological phases have been a major focus in interdisciplinary fields such as condensed matter physics, ultra-cold atoms, and photonic/acoustic simulations over the past decade. However, almost all previous studies have focused on Abelian-type Floquet topological phases. The extension to non-Abelian scenarios is largely unexplored, and the concept of non-Abelian topological invariants is also limited to static cases. In comparison to these extensive studies previously, our paper presents the **first study of Floquet non-Abelian topological phases** emerging from multiple tangled bandgaps. This greatly expands the scope of Floquet topological insulators into the non-Abelian regime, providing exciting opportunities for exploring the rich and uncharted territory of non-equilibrium topological phases, as highlighted in the *Abstract*. Our paper addresses **three key open questions of nonequilibrium topological phases**: (i) the existence and implementation of FNATIs (e.g., concrete models and driving protocols); (ii) the novel bulk-edge correspondence, its description, and physical consequence; and (iii) the identification of topological or dynamical phenomena unique to FNATIs. None of these questions were addressed in previous works, and our comprehensive investigation represents a significant advancement. Our findings should be of great interest to physicists working in the fields of topological physics, non-equilibrium phases of matter and dynamics, and quantum simulations. This is also noted by Reviewer #2, who commented that “*the manuscript studied a very interesting*

problem,” and Reviewer #3, who remarked that “*The dynamics of non-abelian topological systems is a developing field with many questions to address*” and “*Their findings...are also remarkable.*”

(3) Reviewer #1 further mentioned that “*In fact, I find that it has already become a paradigm that people can readily apply to whatever new topological band insulators in static systems and construct its driven version where an anomalous phase shows up.*” We agree with this statement as it primarily focuses on the formal topological characterization of such anomalous phases. Additionally, we would like to emphasize the equal importance of the novel physical effects that these anomalous phases can bring. In the case of the anomalous Floquet Chern insulator, an exotic quantized non-adiabatic pumping effect has been identified (see e.g., Phys. Rev. X **6**, 021013 (2015)). In our work, we have also identified a novel effect, namely, the emergence of interface modes induced by swapped driving, which indicates the intrinsic non-Abelian nature of the underlying quantum dynamics. This effect is counterintuitive and would not occur in Abelian systems because swapped driving (different choices of time frame) only induces a similarity transformation on the Hamiltonian and does not alter the energy spectra on the two sides of the domain wall. Furthermore, via the exact sequence of homotopy groups, as presented in the manuscript, we have rigorously identified this effect and determined the conditions for the emergence of domain-wall states.

We hope that these clarifications will convince Reviewer #1 of the novelty and impact of our work, as well as acknowledge the significant conceptual or technical progress made in this study. In the revised manuscript, we have added several sentences (on page 1) to highlight the impact of our paper.

Reply to Reviewer #2

[Comment] The author studied the PT-symmetric periodically driven 3-band model with non-Abelian topological charge, proposed the realization of anomalous Floquet non-Abelian topological insulator phase, and signatures of interface modes induced by swapped driving as a probe of non-Abelian topology.

[Reply] We thank Reviewer #2 for the careful reading of our manuscript. Following the reviewer’s insightful comments and helpful suggestions, we have revised the manuscript accordingly. Below, we provide a detailed point-by-point response to the reviewer’s comments.

[**Comment**] Overall, the manuscript studied a very interesting problem. However, in the current manuscript the discussion of bulk-boundary correspondence is more like enumerating things without revealing much of a deeper understanding. One should be able to provide some argument about why the boundary states should look at the list in Fig. 3.

[**Reply**] This is a great question. To avoid any confusion, we note that the number of edge states in Fig. 3 is per edge of the system. The term “multifold bulk-edge correspondence” refers to the behavior that a specific quaternion charge of the Floquet Hamiltonian can give rise to multiple distinct edge-state patterns. For example, a k -charge can manifest as either a single edge state in the first gap or two edge states, with one situated in the second gap and the other in the third gap. This multifold bulk-edge correspondence arises from the following two facts:

(1) First, the edge state in each gap is determined by the phase-band closings (or Dirac-point singularities) that occur within that gap during parameter variation (in our case, time evolution). It is well-known that in the theory of topological insulators, each gap closing (Dirac point) leads to a reversal of the mass term, creating an effective “domain-wall” with reversed mass terms on its two sides. The standard Jackiw-Rebbi argument then applies and predicts the emergence of in-gap modes associated with this domain wall [See e.g., *Topological Insulators*, pages 19-22, by Shunqing Shen, Springer, 2012]. In our case, since there are multiple gaps, the edge-state patterns can be determined by considering the phase-band singularities in each bandgap. Furthermore, due to the PT symmetry in our system, each phase-band singularity can be assigned a quaternion charge, denoted by a tilde in the main text (e.g., \tilde{i} , \tilde{j} , etc.). The calculation of the quaternion charge of the phase-band singularity is detailed in the *Methods* section.

(2) Second, the patterns of phase-band singularities determine the quaternion charge of the Floquet Hamiltonian. They are related through Eq. (4) in the main text. The proof of this relation using homotopy group is provided in Appendix A. The strategy is to establish the relationship between the composite path of all the phase-band singularities and the Brillouin zone, where the quaternion charge is defined.

Based on (1) and (2), it becomes evident that the edge-state pattern is linked to the quaternion charge of the Floquet Hamiltonian, and the bulk-edge correspondence exhibits a mul-

tifold nature that follows the multiplication rule ($i^2 = j^2 = k^2 = ijk = -1$; $ij = -ji$; $jk = -kj$; $ik = -ki$) of quaternion invariants. For instance, a charge k Floquet Hamiltonian can arise from either a phase-band singularity of charge \tilde{k} (occurring within the first gap) or from two phase-band singularities of charge \tilde{i} (occurring in the second gap) and \tilde{j} (occurring in the third gap) since $k = \tilde{i}\tilde{j}$. Consequently, there exists either a single edge state in the first gap (in the fifth row and fourth column of Fig. 3) or two edge states situated at the second and third gaps (in the fifth row and fifth column of Fig. 3). A comprehensive list of edge-state patterns and phase-band touchings associated with different quaternion charges is summarized in Fig. 3 of the main text. These patterns obey the multiplication rule of the quaternion group.

In the revised manuscript, we have added discussions about the multifold nature of the bulk-edge correspondence and incorporated the above explanations into the main text.

[Comment] and why the domain wall problem corresponds to the quotient of left side and right side. These are claims in the current version that need some more explanation. I guess some type of Zak phase calculation would apply similar to Ref. 16. But still a good discussion in the Floquet case is needed for better understanding of the manuscript.

[Reply] We appreciate Review #2 for raising these insightful technical questions. They are important in understanding multifold bulk-edge correspondence. In this resubmission, we have added two new sections in the Supplementary Material (SM) to discuss the quotient relation of the domain-wall problem (Appendix D) and the Zak-phase characterization (Appendix E). Correspondingly in the main text, we have also added discussions in section *Multifold bulk-edge correspondence* to address these issues.

1. Regarding the quotient relation in the domain-wall problem:

First, let us clarify that the quotient relation $\Delta q = q_L/q_R$ implies a multifold bulk-domain-wall correspondence. Given specific values of q_L and q_R , the resulting Δq is definite, but may correspond to different patterns of domain-wall states, as also summarized in Fig. 3 of the main text. In fact, the edge-state patterns in Fig. 3 can be viewed as a special case of the domain-wall problem, where the system interfaces with the vacuum. It is the existence of the additional bandgap across the Floquet Brillouin zone that allows for multiple bulk-domain-wall correspondence. To demonstrate the quotient relation, a similar approach

Figure R1: Quotient relation of Floquet non-Abelian topological insulators (FNATIs). (a) Sketch of the domain wall. The left (right) side has bulk quaternion charge q_L (q_R), respectively. The domain wall is characterized by the quotient relation $\Delta q = q_R^{-1}q_L$. (b) Geometric visualization of the quaternion charge as a closed path (loop) in the configuration space M_3 . The change in quaternion charge from q_R (blue) to q_L (orange) is represented by Δq (dotted black), and the concatenated path $q_R\Delta q$ is equivalent to q_L . (c) Relationship between the quaternion charge and the phase-band singularities.

to the static case can be followed. However, due to its multifold nature, more information about the phase-band singularities is required to fully determine the domain-wall states. We can proceed with the following three steps, as sketched in Fig. R1 below.

(1) Between two distinct topological phases, there are gap closings at the domain wall, which correspond to the topological phase transition. This holds true for both static and Floquet cases. It is important to clarify that in Floquet systems, two systems being topologically distinct means that they have different phase-band singularities or edge-state patterns. Notably, in Fig. 3 of the main text, even for the same quaternion charge of the Floquet Hamiltonian, there may be multiple distinct phases due to their different phase-band singularities.

(2) The details of the gap closings, such as their position and times, are encoded in the quotient relation $\Delta q = q_R^{-1}q_L$, as shown in Fig. R1(a,b). This is because the quaternion charge serves as the first homotopy invariant of the configuration space $M_3 = \frac{O(3)}{O(1)^3}$ of PT-symmetric Hamiltonians. Geometrically, the quaternion charge can be visualized as a closed path in M_3 , as depicted in Fig. R1(b). At the domain wall, the quaternion charge undergoes a change from q_R to q_L through gap closings. This change is described by an intermediate

Figure R2: Domain-wall states between two Floquet phases of the same charge j . (a) A domain wall between $q_L = j$ (with singularity \tilde{j}) and $q_R = j$ (with singularity $\tilde{k}\tilde{i}$) possesses three domain-wall states (one in each gap), as per $\Delta q = 1 = (\tilde{i}\tilde{k})\tilde{j}$. (b) A domain wall between $q_L = j$ (with singularity \tilde{j}) and $q_R = j$ (with singularity \tilde{j}) does not possess any domain-wall state. The domain wall is set at the 26-th site.

path (the dotted black loop) denoted as Δq . In homotopy language, the concatenation of the two paths $q_R \Delta q$, i.e., first following q_R and then Δq , yields the path q_L : $q_L = q_R \Delta q$. Then we have $\Delta q = q_R^{-1} q_L$. Intuitively, Δq represents first following the inverse of q_R (to cancel the q_R path) and then following q_L . Furthermore, in Floquet systems, the quaternion charge on each side is related to the phase-band singularities through Eq. (4) of the main text, as sketched in Fig. R1(c). Thus, we have $\Delta q = q_R^{-1} q_L = (\tilde{q}_{R1} \tilde{q}_{R2} \dots)^{-1} (\tilde{q}_{L1} \tilde{q}_{L2} \dots)$.

(3) The gap closings recorded in Δq of step (2) give rise to domain-wall states via the Jackiw-Rebbi argument. The number of gap closings and their positions (i.e., in which bandgap) correspond to the number of edge states in that gap. In other words, whenever there is a gap closing, there appears a domain-wall state. To compare with the static case and showcase the multifold nature, we take an example domain wall with the same charge $q_L = q_R = j$, as shown in Fig. R2. $\Delta q = 1$. The domain wall between $q_L = j$ (with singularity \tilde{j}) and $q_R = j$ (with singularity $\tilde{k}\tilde{i}$) possesses three domain-wall states as $\Delta q = (\tilde{i}\tilde{k})\tilde{j}$ in Fig. R2(a). While the domain wall between $q_L = j$ (with singularity \tilde{j}) and $q_R = j$ (with singularity \tilde{j}) possesses no domain-wall state in Fig. R2(b). These two different patterns of domain-wall states correspond to the same $\Delta q = 1$.

From (1)(2)(3), we conclude that the domain-wall states are described by the quotient relation $\Delta q = q_R^{-1} q_L$. Different from the static case, the specific gap closings across the domain wall and the patterns of domain-wall states are fully determined by the phase-band singularities. This multifold bulk-domain-wall correspondence is summarized in Fig. 3 of the main

text.

2.Regarding the Zak-phase characterization:

In the added Appendix E, we have performed a detailed calculation of the Zak phase, the results are listed in Table 1 below.

Table 1: Zak phase of the quasienergy band

q	1	$\{\pm i\}$	$\{\pm j\}$	$\{\pm k\}$	-1
Third band	0	π	π	0	0
Second band	0	π	0	π	0
First band	0	0	π	π	0

The Zak phase alone is inadequate for predicting edge states or domain-wall states for two reasons. (1) The Zak phase is assigned to each individual band and can only take two values 0 or π . Within the same conjugacy class (e.g., i and $-i$), the Zak phase for each band remains the same. It is worth noting that in the static case [See Fig. 4(c) of Ref. [16]], the Zak phase is also assigned other values, such as $-\pi$ and 2π for the purpose of differentiating between two conjugate elements within the same conjugacy class. In actual calculations, $\pm\pi$ corresponds to different gauge choices of the eigenfunctions and is considered equivalent in our notation. The constraint that the sum of Zak phases for all three bands equals 0 (mod 2π) leads to four possible Zak phase patterns as in Table 1. Since there are five conjugacy classes in the quaternion group, the charge of -1 (which has the same Zak phases as charge 1) falls outside the scope of the Zak phase description. (2) The Zak phase is defined for the Floquet Hamiltonian, while the system's topology is fully encoded in the time-evolution operator or phase-band singularities. As discussed previously, the bulk-edge or bulk-domain-wall correspondence is multifold. Different patterns of edge/domain-wall states can possess the same quaternion charge (or Zak phase pattern). For both reasons above, we conclude that the Zak phase is insufficient for predicting the edge/domain-wall states. A comprehensive understanding of the system's topology requires accounting for the phase-band singularities.

[Comment] For this domain wall problem, I have a minor technical question: maybe I have some misunderstanding but why the quotient is -1 for the swapped driving case but the interface states look like j in Fig. 3?

[Reply] In Fig. 4(b), there are two domain-wall states (located near $\pm\frac{\pi}{T}$) within the third gap because the domain wall is described by the relation $-1 = \tilde{j}^2$. They are close to each other due to the periodicity of the quasienergy. In the revised main text, we have clarified this point. This scenario contrasts with the case of charge j , with only a *single* domain-wall state in the third gap.

[Comment] Furthermore, usually for the topology of a fixed number of bands such as the multi-gap topology here, the bulk-boundary correspondence is sensitive to boundary conditions since on boundaries/domain walls there can be more degrees of freedom. This is what happened in e.g., delicate topology. I am wondering how general the bulk boundary correspondence is upon changing boundary condition or decorating boundaries/domain walls and how stable it is about the appearance of domain wall states. I believe here some precise statements need to be made to clarify the stability of the bulk-boundary correspondence especially for the intrinsically non-Abelian part.

[Reply] We appreciate the reviewer for raising this intriguing question. We seek to provide further clarification on the nature of multi-gap topology, both in the static and Floquet cases. The multi-gap topology should be regarded as a form of fragile topology, which lies beyond the Altland-Zirnbauer tenfold way. Fragile topology is contingent on the number of bands, and the band topology may be trivialized by coupling with additional trivial bands. In the context of multiple tangled bandgaps, the fragile topology was defined via repartitioning of energy bands, leading to a reduction of topological invariants, as addressed in the literature. [See e.g., page 11 of Ref. [10].] In the case of three bands, we may have either a $1 + 1 + 1$ or a $2 + 1$ band partition. For the former, the first homotopy group of the flag manifold yields the quaternion group, which is non-Abelian and relevant to our paper. While for the latter, the flag manifold is $\mathbb{R}P^2$, with the first and second homotopy groups being Abelian, given by \mathbb{Z}_2 (corresponding to Dirac string) and \mathbb{Z} (corresponding to Euler class), respectively. It is worth noting that, akin to strong topological phases such as the Chern insulator or time-reversal invariant topological insulator, fragile topology also imposes an obstruction to an atomic limit. The above scenario is different from the delicate topology with Hopf or returning Thouless pump invariant, which has localized but multicellular Wannier representations [see e.g., A. Nelson, T. Neupert, T. Bzdušek, and A. Alexandradinata, Phys. Rev. Lett. **126**, 216404 (2021); Phys. Rev. B **106**, 075124 (2022)].

To address the stability of the edge states and domain-wall states against various boundary

conditions, disorders, and domain-wall decorations, we have included a new section (Appendix F) in the SM. Our numeric indicates that the edge states/domain-wall states remain stable in the presence of disorder [See Fig. S5]. And the domain-wall states are also robust against different domain-wall decorations [See Fig. S6]. It is worth noting that although additional *trivial bound states* may arise due to the decorations, the nontrivial domain-wall states, which originate from the non-Abelian topology and are governed by the phase-band singularities, stay intact and remain robust [See Fig. S7].

[Comment] In summary, I believe the current manuscript can be improved to meet the standard of Nature Communication and a more transparent and thorough understanding of the phenomena discussed in the current manuscript is needed.

[Reply] We appreciate the valuable feedback from Reviewer #2 and have diligently revised and enhanced our manuscript, aiming to provide a more transparent and comprehensive understanding of the phenomena discussed. We hope that the current version of the manuscript adequately addresses the reviewer’s questions.

Reply to Reviewer #3

[Comment] This work proposes a Floquet topological insulator model characterized by non-abelian charge. They show the bulk-edge correspondence can be derived from the phase-band singularities, and there is an “anomalous” phase in which the bulk topological invariant is trivial but the edge mode is protected. They also found an novel interface effect as a result of the non-Abelian topology.

The dynamics of non-abelian topological systems is a developing field with many open questions to address. This work provides the first Floquet model with non-Abelian topology and thoroughly study the bulk-edge correspondence to the best of my knowledge. Their findings of the “anomalous” edge state and the interface modes are also remarkable. These prediction might be tested in future experiments. So far as I see, they provide enough analytical and numerical calculations to justify their statements. Therefore, I would like to recommend its publication on Nat Com.

[Reply] We appreciate Reviewer #3 for recognizing and providing a positive assessment of our work, as well as the recommendation for its publication in *Nature Communications*.

[Comment] There are two questions that the authors might want to address in a modified manuscript. 1) Can the authors comment on the possibilities of generalizing their work to higher spatial dimensions, like 2D? 2) Can the authors comment on the possibilities of generalizing their work to Floquet Chern insulators?

[Reply] We thank Reviewer #3 for the questions regarding the potential generalizations of our work. In the revised manuscript, we have included additional comments in the *Concluding remarks* section to address these possible extensions. We hope they should open up new avenues for exploration and could provide further insights into the interplay between non-Abelian topology and Floquet systems.

(1) Generalization to higher dimensions: It would be intriguing to extend our work to higher dimensions. The quaternion charge we considered in the manuscript corresponds to the first homotopy group of the complete flag manifold $M_3 = O(3)/O(1)^3$, which goes beyond the tenfold way and captures the fragile topology associated with multiple tangled gaps. Generalizing to higher dimensions involves considering higher homotopy groups of the flag manifold associated with different band partitions. Here are some potential extensions: (i) 2D Floquet topological Euler phase with band nodes: In the presence of Floquet driving, band nodes can be fully annihilated through reciprocal braiding, resulting in an anomalous Dirac string phase, as recently introduced in Ref. [14]. (ii) 3D Floquet nodal-line metals characterized by non-Abelian frame charges: With Floquet driving, additional nodal lines can emerge across the quasienergy zone edge. It would be of great interest to explore the interplay between these non-Abelian nodal lines and their associated physical effects. Specifically, we are curious about the constraints on possible nodal-line configurations imposed by the non-Abelian charges in Floquet systems. (iii) Relaxing PT symmetry (i.e., the real valuedness of the Hamiltonian) and exploring non-Hermitian systems: By allowing complex eigenenergies, non-Hermitian systems associated with complex flag manifolds exhibit non-Abelian properties. Previous works [See e.g., Ref. [11,53-55]] have revealed the non-Abelian nature of non-Hermitian bands and the braiding of exceptional points /lines in 2D/3D due to the nonstandard gap conditions. How these non-Abelian features interplay with Floquet driving is a natural question to explore. (iv) Generalizing to higher-band cases: An immediate extension is to consider the four-band multi-gap topological insulator characterized by the generalized quaternion group Q_{16} .

(2) Generalization to Floquet Chern insulator (FCI): FCI is a well-studied non-equilibrium topological phase that exhibits Abelian (\mathbb{Z}) topological invariants. In Ref. [42], the appearance of anomalous edge states despite trivial bulk bands, were discovered and connected to the micromotion and phase-band singularities [See e.g., Ref. [46]]. While there are similarities in the anomalous behaviors and the phase-band characterization between FCI and our Floquet non-Abelian topological insulator (FNATI), we note their key differences. *(i)* Symmetry: FNATI requires PT symmetry, ensuring that the Bloch wavefunctions are real-valued. On the other hand, FCI does not require any specific symmetry. With PT symmetry present, FCI would be trivial, and its topological invariants are zero. *(ii)* Classification: FCI falls within the tenfold way as listed in Refs. [39,40], while FNATI in our work goes beyond the tenfold way and belongs to the realm of fragile topology [See also our response to Review #2]. *(iii)* Gap condition: In FCI, each bandgap is considered independently, and the topological invariant is defined for each individual gap or band. In contrast, FNATI exhibits tangled bandgaps. The topological invariant governs the entire band structure and the edge-state patterns with multifold bulk-edge correspondence, as listed in Fig. 3 of the main text. With these key distinctions, the features of FNATI are not captured by the traditional framework of FCI.

Summary of main changes

We have revised the main text and the Supplementary material following all three reviewers' comments/suggestions. The changes have been marked in red in the revised manuscript. Here is a list of the main changes.

1. We have added three new sections, including one new table (Table S1) and five new figures (Fig. S3-S7), in the Supplemental Material to address the quotient relation of domain-wall systems (Appendix D), the Zak phase of the quasienergy band (Appendix E), and the stability of the edge/domain-wall states (Appendix F).
2. In the third paragraph (page 1), we have added several sentences to stress the novelty and impact of our paper.
3. In section *Multifold bulk-edge correspondence*, we have provided explanations about the multifold bulk-edge correspondence (at the end of the first paragraph, page 3).
4. In section *Multifold bulk-edge correspondence*, we have added discussions about the domain-wall problem (at the end of this section, page 4).
5. We have added a sentence in the caption of Fig. 3 to explicitly reference the domain-wall problem (page 4).
6. In section *Interface modes induced by swapped driving*, we have clarified the domain-wall states in Fig. 4(b) (in the first paragraph, page 5).
7. In section *Concluding remarks*, we have reorganized the second paragraph and added discussions on the potential extensions of our work (page 5).
8. Table I has been expanded to incorporate additional data used in the paper.
9. We have added 4 new references (Refs. [51,53-55]) and updated Refs. [11,58].

REVIEWERS' COMMENTS

Reviewer #1 (Remarks to the Author):

I appreciate the effort that the authors have put in revising the manuscript according to the comments raised by all referees. While I stand by my previous opinion that the current work may not represent the most significant progress in our understanding of Floquet topological phases, from a practical perspective, the particular model and phenomena proposed in this work may be of interests to a fairly broad audience working in this field. From a more theoretical perspective, I still believe that the results reported in this work are combinations of several elements that are already known. While I agree that such a combination sometimes leads to phenomena that do not exist in the individual settings, I am not sure I would call the phenomena reported in this work extremely intriguing. Overall, I can support publication of this paper, albeit not with the greatest enthusiasm.

Reviewer #2 (Remarks to the Author):

The authors now have highlighted the reasoning behind the multifold bulk-edge correspondence where the quaternion charge of the Floquet Hamiltonian can correspond to different edge state patterns. The authors elegantly relate these properties to the different possible patterns of phase-band singularities. These are nice concrete results for the bulk-edge correspondence of the Floquet system with non-Abelian topology which have experimental implication. With the improved manuscript, the authors have addressed my concern and I would like to recommend the publication on Nat. Comm.

Reviewer #3 (Remarks to the Author):

I am satisfied with the authors' response and the modified manuscript. The newly added sections clarified some important questions brought by the other referees and significantly improved the quality of this paper by providing deeper insights to their model. They convinced me of the novelty and rigorosity of this paper. Therefore, I recommend its publication.